# Exploring the Habitability of Venus: Conceptual Design of a Small Atmospheric Probe

Pol Ribes-Pleguezuelo *,† , Bruno Delacourt † , Mika K. G. Holmberg † , Elisabetta Iorfida † , Philipp Reiss † , Guillermo Salinas † and Agnieszka Suliga †

Research Fellow Study Group, European Space Research and Technology Centre, European Space Agency, Keplerlaan 1, 2201 AZ Noordwijk, The Netherlands; bruno.delacourt@esa.int (B.D.); mika.holmberg@pm.me (M.K.G.H.); elisabetta.iorfida@esa.int (E.I.); philipp.reiss@esa.int (P.R.); guillermo.salinas@esa.int (G.S.); agnieszka.suliga@esa.int (A.S.)

\* Correspondence: pol.ribes@esa.int

† These authors contributed equally to this work.

**Abstract:** The possible presence of life in the atmosphere of Venus has been debated frequently over the last 60 years. The discussion was recently reignited by the possible detection of phosphine ($PH_3$), but several other chemicals potentially relevant for life processes are also found in the middle atmosphere. Moreover, the reasons for the heterogeneous ultraviolet (UV) absorption between 320 and 400 nm in the altitude range ~40–70 km are still not well understood. These aspects could be further studied in-situ by UV Raman and fluorescence instruments. Here, the conceptual design of a small balloon probe (<20 kg) is presented, including a science payload comprising a UV laser, spectrometer, and a telescope. The goal of the proposed mission is to analyse the absorption of UV light in Venus' atmosphere, to study the atmospheric composition, and to verify the possible presence of biomarkers. Current state-of-the-art technologies would allow a more cost-efficient and easy to develop mission, as compared to previous Venus probes. This article is focused on the scientific instrumentation, as well as on the mass and power budgets required to realise the proposed mission.

**Keywords:** Venus' atmosphere; Raman spectroscopy; ultraviolet; biomarker; phosphine; fluorescence; mission concept; nanosatellites; EnVision

## 1. Introduction

### 1.1. Mission Background

Acidic cloud aerosols in the atmosphere of Venus, located in a region with Earth-like pressures and temperatures, could potentially host microbial life (e.g., [1–4]). Several chemical components relevant for the astrobiology and habitability of Venus are found in its atmosphere, such as $H_2$, $N_2$, CO, $CO_2$, $H_2S$, $SO_2$, $SO_4$, OCS, $NH_4Cl$, $NH_2COOHN_4$, and $H_2O$ [1,3–6]. These components could easily lead to consecutive creation of amino acids [5], which is the very first step for further biological processes [3,7]. In addition, from a geocentric understanding of biogenic elements, a sufficient abundance of C, H, O, N, S, and P is needed in order to sustain microbial extremophiles in the atmosphere of Venus [2,4]. The elements C, H, O, N, and S are abundant in the middle atmosphere, but P has only been detected once [8], and more recently in the form of phosphine ($PH_3$) [9]. The validity of both observations is still debated.

UV Raman devices are a key tool for detecting the presence of elements relevant for the astrobiology and habitability of Venus' atmosphere [6]. Furthermore, the highly heterogeneous absorption between 320 and 400 nm in the altitude range ~40–70 km [10] is a phenomenon that can potentially be resolved with a single optical payload. Possible explanations for these observations are the presence of $FeCl_3$ and $SO_2$, or the absorption of solar energy by microorganisms present in the cloud layers of Venus [3].

An in-situ analysis of the atmosphere, as proposed here, can be realised using combined Raman spectroscopy and a laser-induced fluorescence (LIF) device. Measurements from such instruments would improve our understanding of the composition and behaviour of the planet's atmosphere. The need for spectroscopy experiments to study the clouds of Venus has also been stressed by earlier mission proposals [3]. Furthermore, performing atmospheric spectroscopy with balloon probes was suggested by the Venus Science and Technology Definition Team in 2009 [11]. A recent example is the Russian Venera-D mission [12], with an instrument package including an imaging camera, a gas chromatography–mass spectrometer, an alpha P-X-spectrometer, a gamma spectrometer, a laser spectrometer, a hazemeter, and a seismic detector [13]. The spectrometer uses a multi-channel tunable diode laser spectrometer. Furthermore, NASA started developing ideas for a CubeSat mission and descending probes for studying Venus' atmosphere that incorporate similar instruments [14,15].

The goal of the work presented here is to identify the specific science to be conducted in-situ, to propose an optical payload baseline design for an atmospheric probe, and to present a rough estimation of the system design budget for the realisation of the mission.

### 1.2. Missions to the Atmosphere of Venus

The history of Venus exploration started in the early 1960s with the first Soviet missions performing flybys and eventually landings (in the 1970s). Although these first missions were not as successful as planned, they paved the way for future Soviet missions, which could finally analyse the planet's atmosphere. The first successful atmospheric mission, the Venera 4 probe (383 kg), was carried out by the Soviet Union in 1967 [16]. The mission already included a UV spectrometer that helped to analyse its atmospheric composition for the very first time. In 1985, the Vega balloon probes successfully studied the atmosphere of Venus at an altitude of around 50–55 km. The use of a helium balloon, deployed by the descending probe, helped scientists to obtain a much longer analysis time and region of study, due to the dense atmosphere composition and presence of strong winds [17]. Earlier, in 1978, NASA had sent the Pioneer Venus multiprobe (around 600 kg), which performed an analysis of the atmosphere with different independent probes and instruments (amongst others, a mass spectrometer and gas chromatograph to measure atmosphere composition, a cloud particle size spectrometer, and temperature and pressure sensors) [18].

In 2005, ESA launched its first mission to Venus, the Venus Express. The spacecraft spent more than eight years studying Venus and its atmosphere, mainly focusing on high-resolution imagery and spectroscopy. Venus Express was a European flagship mission that inspired many other mission concepts. It was followed by Akatsuki, a successful Japanese Venus mission [19]. However, from the 1960s to the 1980s, no further landing probes were sent into the planet's atmosphere. Nowadays, the technical advances in Raman spectrometry allow for more accurate and extensive analysis than the missions four decades ago. Moreover, new technological developments allow the use of a considerably smaller and lighter platform, which can result in a significantly more cost-effective mission.

In June 2021, both ESA [20] and NASA [21] announced the selection of three new missions to explore Venus. ESA is planning to launch EnVision in 2032 [22] to address three main aspects of our neighbour planet: its history (evolution of surface and interior), its geological activity, and its climate. NASA is expecting to launch another two proposed missions around 2028–2030. DAVINCI+ will be an atmospheric mission with a descent sphere to understand the formation and evolution of the planet, and to also determine whether Venus has ever had an ocean. VERITAS (SAR mission like EnVision) aims at mapping the geological history and development of the planet. The newly selected missions are all unique and complementary.

### 1.3. Space Raman Devices

Raman spectroscopy is an interesting technology for planetary science, since it allows for in-situ analysis of samples thanks to the inelastic scattering of light. Fundamentally,

the instrument only requires a narrow light excitation source, usually realised by a laser, and a spectrometer to analyse the Raman light shift (Figure 1). The latter provides accurate information about the vibrational and rotational molecule levels of the illuminated sample. Raman spectroscopy can measure samples at a distance of up to 200 m and is therefore useful for long-range atmospheric measurements [23].

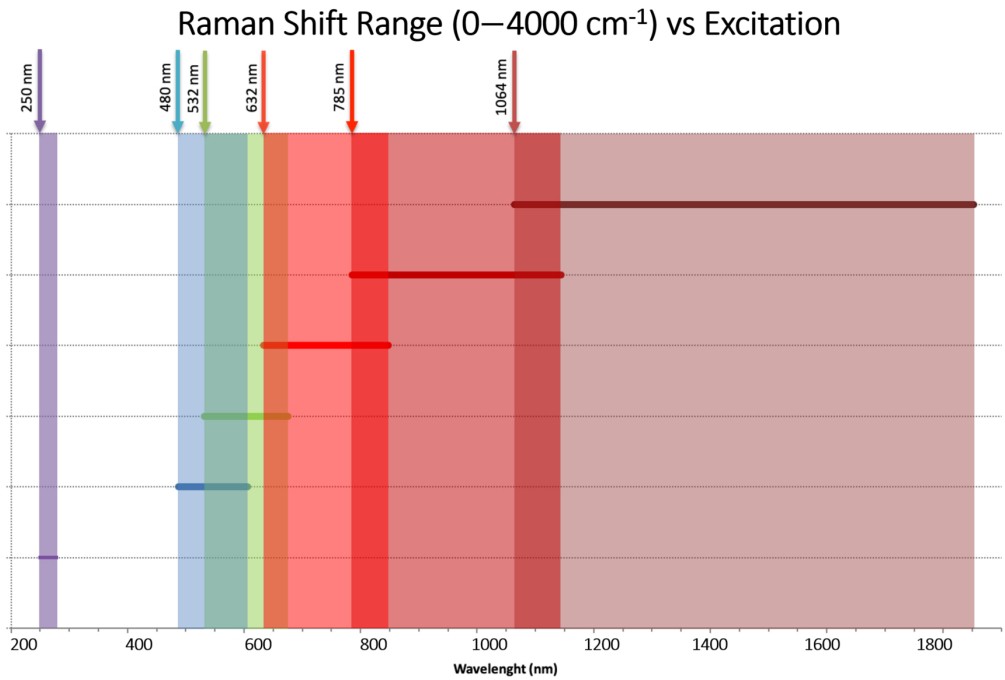

**Figure 1.** Raman shift range versus laser excitation. Raman shift or excitation of bounced light into the target material depends on the excitation source/laser. For a certain wavelength input, a different output shifted wavelength is obtained with different peaks. This shift can be used to identify the analysed molecules. Reprinted from [24].

Raman spectroscopy allows for precise identification of both mineral and organic phases down to sub-picogram levels [25] without disturbing the analysed sample. This analysis can be conducted in any state in which the sample is found (solid, liquid, or gas) and with micrometre resolution. In addition, Raman spectroscopy is a very fast technique that allows measurement within a few seconds. Along with this, the data products are very simple, so the memory consumption and processing requirements are minimal. This technique can be applied both in contact mode, bringing the signal through optics and fibres to the spectrometer, as well as remotely, from a few centimetres to tens of metres. Contact measurements can be done using a continuous laser and in low light conditions, while remote acquisition is carried out using a pulsed laser.

Over the last decades, several research groups have been developing compact Raman systems that are capable of operating in extreme space and planetary environments. In 2006, a team at the University of Hawaii implemented several changes to standard Raman spectroscopy devices to be able to create a small portable system for outdoor operation [26]. Already planning for space exploration, they designed a device with a 532 nm laser, an energy of 35 mJ, and 8 ns pulses. The team reported that they were successful in their goals, thanks to the use of volume holographic transmission gratings with a resolution of 9 cm$^{-1}$. The device used a Nd:YAG laser with second harmonic generation (SHG) to obtain the 532 nm emission. The incoming light from the sample was later collected using a Makusutov Cassegrain 127 mm telescope [26].

Furthermore, several planetary missions are currently using compact Raman devices to perform research, the most recent example being NASA's Mars 2020 mission [27]. Its rover Perseverance uses the SuperCam instrument to perform Raman spectroscopy at up to

7 m distance. SuperCam includes, in addition to Raman, a suite of measurement techniques such as Laser-Induced Breakdown Spectroscopy (LIBS), Time-Resolved Fluorescence (TRF) spectroscopy, Visible and InfraRed (VISIR) reflectance spectroscopy, and high-resolution visual imaging. The mission also uses a robotic arm that includes the Scanning Habitable Environments with Raman & Luminescence for Organics & Chemicals (SHERLOC) Raman device, to perform proximity measurements with a 248.6 nm laser emission [27]. Similarly, ESA is planning to launch an instrument in the frame of the ExoMars mission in 2022, which will also have Raman capabilities [28].

A Raman instrument specifically designed for the investigation of Venus' atmosphere would be very suitable for the detection of amino acids and their precursors. There is already extensive experience in the use of this technology in the determination and differentiation of amino acids in various environments, whether deposited on inorganic surfaces, in solution, or in aerosols. In addition, the Raman spectroscopy technique is capable of distinguishing types of amino acid by generating a unique fingerprint based on its molecular composition. Moreover, recent publications [9] state that phosphine in Venus' atmosphere could be produced by certain types of bacteria. This can also be easily studied with a Raman technique. The Raman shift range between 500 and 1700 $cm^{-1}$ provides a sufficient signal for the precise determination of target amino acids (Figure 1). In particular, performing distance Raman and fluorescence analysis using a pulsed UV excitation source can help to determine the presence of organics, amino acids, or other more complex biomarker molecules in Venus' atmosphere (Figure 2).

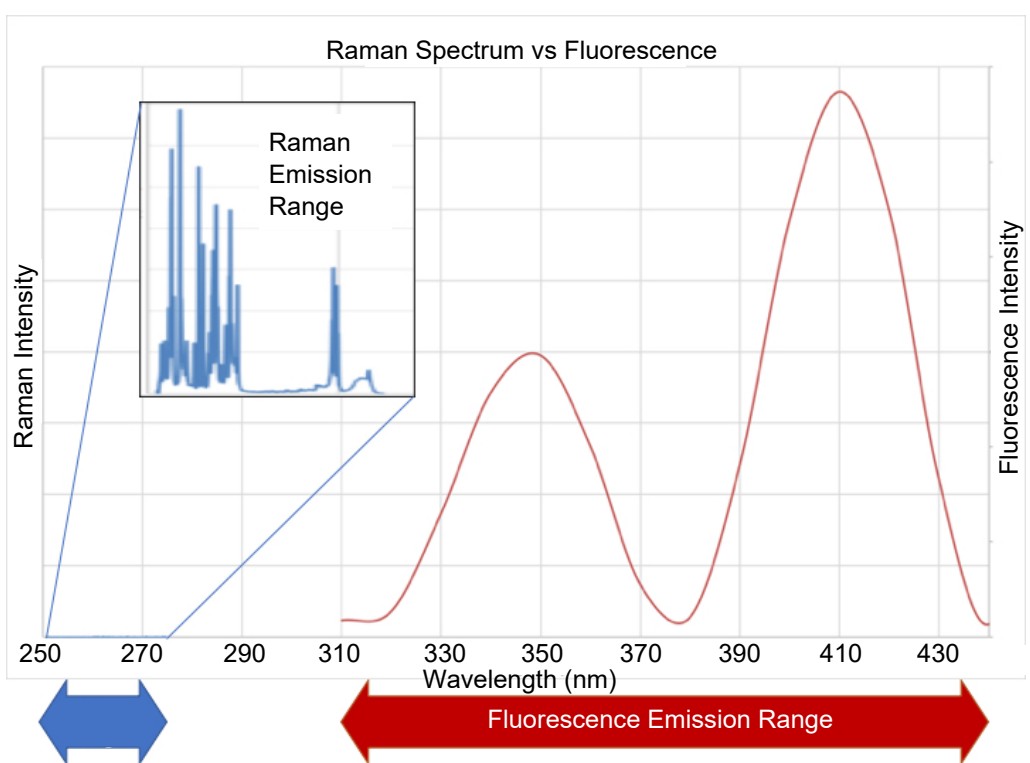

**Figure 2.** Raman and fluorescence emission ranges for a 250 nm excitation. Reprinted from [24].

### 1.4. Venus Atmospheric Conditions

One major constraint of performing Raman spectroscopy in Venus' atmosphere is the presence of extreme and challenging conditions under which the entire probe will have to operate. The investigation of existing chemicals and possible amino acids requires the probe to operate in highly variable pressure and temperature regimes, depending on the altitude [6,29].

The atmosphere of Venus is commonly divided into the lower, middle, and upper atmosphere. The lower atmosphere commonly refers to the region below ∼65 km above the surface, while the middle atmosphere ranges from ∼65 to ∼95 km and includes the stratosphere and the mesosphere [30]. The upper atmosphere is the region above ∼95 km and includes the thermosphere and the exosphere [30]. The atmospheric region of interest for the present study is between 40 and 70 km altitude. Here, the temperature and pressure varies from approximately 225–400 K and 10–1000 mbar, respectively [6,30,31]. Figure 3 shows the density and temperature versus altitude and pressure, as obtained from the Venus International Reference Atmosphere (VIRA) model. VIRA was originally published in 1985 [32] and has since been updated continuously as new data and important findings were provided by later Venus missions, such as Vega 1, Vega 2, and Venus Express, and by new ground based observations. Venus has a dense cloud layer that ranges from around 48 to 65 km, above which the particle concentration falls off with a scale height of around 3 km [30]. The different cloud layers and their respective altitude ranges are also shown in Figure 3. The clouds are primarily composed of liquid droplets of sulphuric acid ($H_2SO_4$) [30].

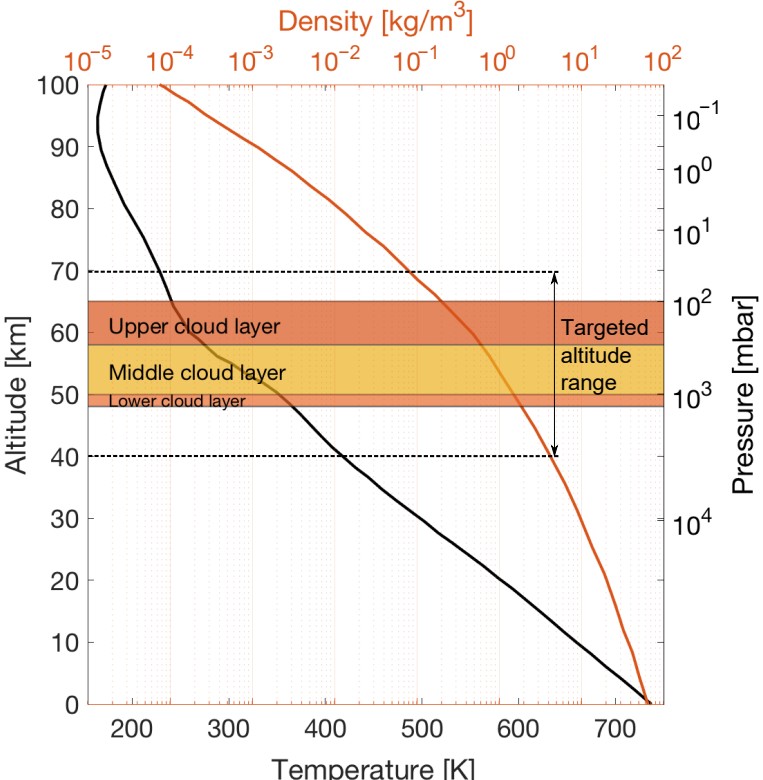

**Figure 3.** Atmospheric density (red line) and temperature (black line) versus altitude and pressure, based on the VIRA model. Figure adapted from Figure 1 in [33].

Recent studies [34–36] combine the analysis of zonal winds at Venus' cloud-top from observations in space and on the ground. A clear common wind profile, between 50 degrees south and 50 degrees north (midlatitude region) with an average value of 100 m/s, can be seen in Figure 4. At around 50 degrees, a smooth jet of, at most, 10 m/s is noticeable in both hemispheres, whereas at higher latitudes there is a steady and steep decrease. For the purpose of the present study, an average zonal wind velocity of 100 m/s can be considered. These strong winds make balloon exploration very favourable, since the probe would be able to travel horizontally, covering vast atmospheric areas [37]. The Soviet Vega 1 and 2 missions in 1985 had already taken advantage of this, and a similar approach was also proposed to ESA in 2013 in response to the Cosmic Vision Call for Ideas [38].

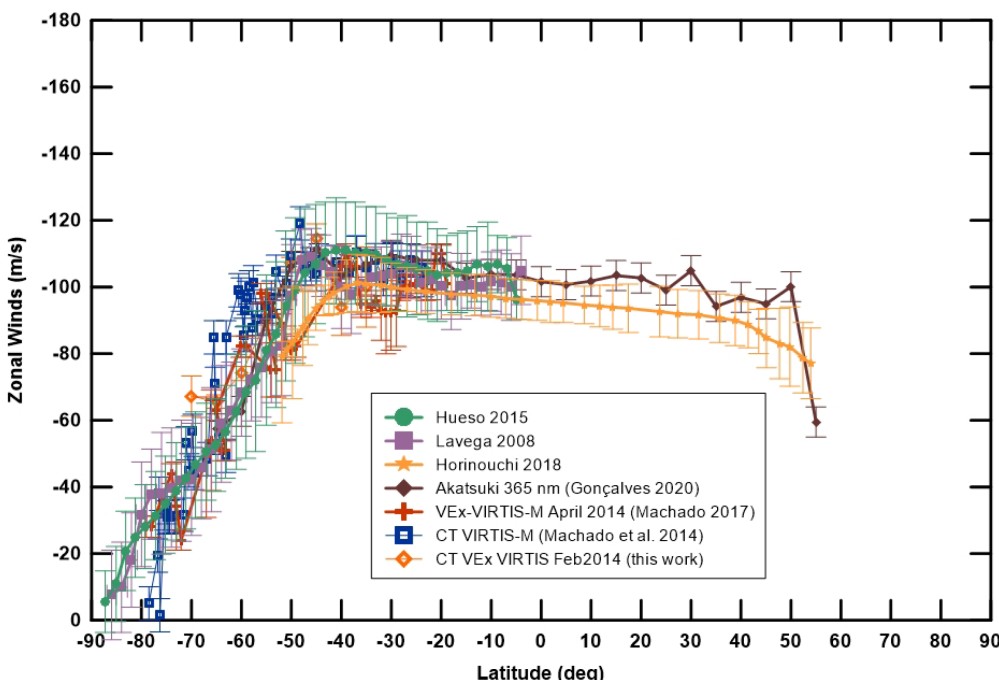

**Figure 4.** Space and ground observation results of zonal wind latitudinal profiles at Venus' cloud-top (70 km). Reprinted from [36].

The atmosphere of Venus is composed of 96.5% $CO_2$, 3.5% $N_2$, and several trace gases, such as $SO_2$, Ar, $H_2O$, CO, He, and Ne. The individual abundance in the region of interest is listed in Table 1. HCl, Kr, HF, and Xe have also been observed, but with an abundance between 1ppb and 1 ppm [30]. A recent detection of $PH_3$, with an abundance of around 20 ppb above 48 km, sparked new interest in the search for life on Venus [9]. Investigation of various chemical processes that could be the source of the $PH_3$ suggested that no currently known process could explain the detection. The presence of $PH_3$ was suggested to be due to some unknown photochemistry or geochemistry process, or due to the presence of life [9]. The planet-averaged $PH_3$ abundance was later revised to 1–4 ppb, with peaks at 5–10 ppb [39]. However, the presence of $PH_3$ in the atmosphere of Venus is controversial and has been questioned by recent studies. For example, [40,41] showed that the data processing method used by [9] can result in spurious lines, including the spectral feature of $PH_3$.

**Table 1.** Composition of the atmosphere of Venus. The given abundances are obtained from [30,42,43].

| Gas | Abundance |
|---|---|
| Carbon Dioxide ($CO_2$) | 96.5% |
| Nitrogen ($N_2$) | 3.5% |
| Sulfur Dioxide ($SO_2$) | 10–260 ppm |
| Argon (Ar) | 20–200 ppm |
| Water Vapour ($H_2O$) | 20–30 ppm |
| Carbon Monoxide (CO) | 17–40 ppm |
| Helium (He) | 12–17 ppm |
| Neon (Ne) | 5–7 ppm |

## 2. Proposed Mission Scenario

### 2.1. Mission Objectives

The overall scientific objective of the proposed mission is to characterise the abundance of biomarkers (phosphine, amino acids, and other constituents) in the atmosphere of Venus.

To ensure the achievement of this objective, the following top-level mission requirements are derived:

- TR1: The mission shall be able to determine the atmospheric conditions and composition, particularly the presence and abundance of phosphine and other relevant biomarkers in the atmosphere of Venus at altitudes between 40 and 70 km.
- TR2: The UV absorption characteristics shall be analysed in different atmospheric regions of Venus.
- TR3: The mission shall be feasible within a budget of a maximum of 50 million Euro (ESA S-class mission).
- TR4: The mission shall achieve flight readiness level in less than 4 years.
- TR5: Commercial off-the-shelf (COTS) components should be used wherever possible, to demonstrate their suitability for space exploration missions.
- TR6: The probe's lifetime in the atmosphere of Venus shall be a minimum of two weeks.

The current state-of-the-art technologies required to satisfy the above requirements allow for a mission with moderate cost and fast development [37], as compared to previous Venus probes. A small probe that makes use of COTS components for nanosatellites is presented here, taking advantage of the existing standardisation in this field and the wide availability of components. The probe, comprising the scientific payload and all required subsystems for survival, shall be carried on an atmospheric balloon to enable long-term observations in the atmosphere.

*2.2. Flight Opportunities*

The proposed probe shall be launched as a secondary payload together with a primary mission, which will also deliver it to the atmosphere of Venus. The main reason for this approach is to reduce cost and development time. Besides the shared launch opportunity, the mission shall use the existing communication/relay network established for the primary mission. A potential joint flight opportunity would be the ESA mission EnVision. The proposed baseline launch date for EnVision is 2032, with a back-up date in 2033. EnVision is planned to be launched with Ariane 62 into a Highly Elliptical Orbit (HEO). Once in HEO, two escape sequence manoeuvres will take the spacecraft into the interplanetary transfer trajectory, a phase that will last 134 days. After the Venus Orbit Insertion (VOI), the spacecraft will go through an aerobraking phase of around 25 months, which will lower the apoapsis from 250,000 to 470 km while holding the periapsis at 220 or 250 km. With this suggested mission timeline, the total EnVision mission duration will range from 5.1 to 6.3 years, depending on the launch date [44].

As a secondary payload, the proposed Venus probe will be able to use EnVision's Earth communication link, thus reducing the antenna requirements. Furthermore, the launch costs will be shared.

### 3. Probe Design

Figures 5 and 6 show the preliminary design and configuration of the probe. The instrument section in the lower compartment encloses a volume of approximately $200 \times 300 \times 100$ mm and contains the UV laser, spectrometer, telescope, and camera. The electronics compartment with dimensions $200 \times 300 \times 200$ mm contains the battery (BAT), the power control and distribution unit (PCDU), the communications unit (COM) with the S-band antenna module, and the onboard computer (OBC). The probe also includes a number of additional sensors to measure the atmospheric ambient conditions, such as temperature, pressure, humidity, and wind speed. At the sides of the electronics section, the parachutes and the balloon are stowed, together with the deployable antennas. The heat shield at the bottom of the probe protects the probe during entry and will be ejected after arriving at the altitude targeted for balloon deployment.

Table 2 shows the average system power and mass budgets. Regarding the power and mass requirements of the science payload, the NASA SHERLOC instrument [45,46] serves as an example.

The balloon consists of the floating mass, which is comprised of the balloon fabric, the buoyant gas, and the gondola, which carries the scientific payload. The floating mass is contained in the entry capsule together with the parachute. Additionally, the capsule includes the inflation mechanism and balloon support structure. The mass breakdown for the entry capsule is not accounted for in Table 2, however, for the 9 kg payload, a maximum of 115 kg would be needed to bring the science probe into the atmosphere [47] (compared to the planetary capsules designed for missions of a similar size [48]). Given the new technological material developments in the area of lighter-than-air systems, it is very likely to even further reduce the mass budget [49].

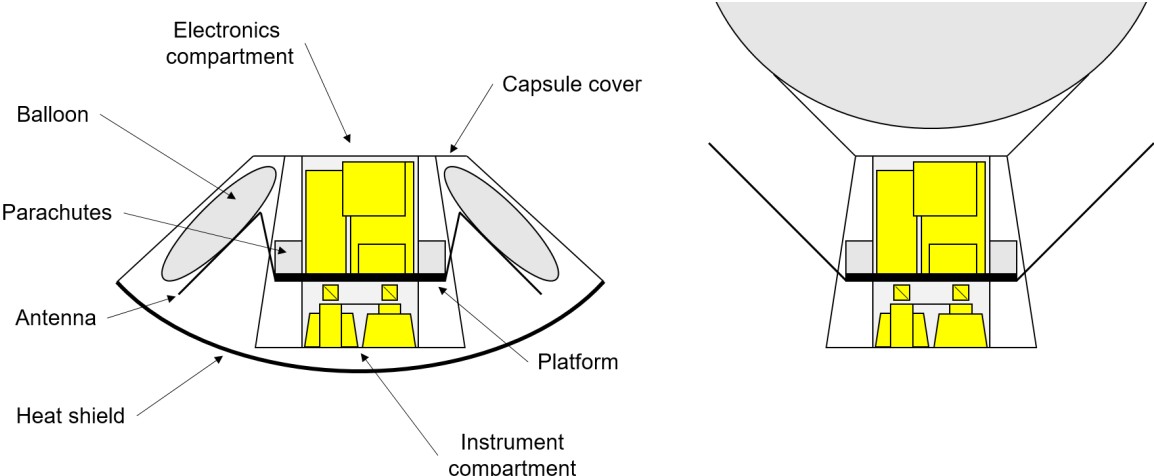

**Figure 5.** Conceptual design of the proposed Venus probe in stowed (left, during cruise and entry) and deployed (right, during operations in the atmosphere) configurations.

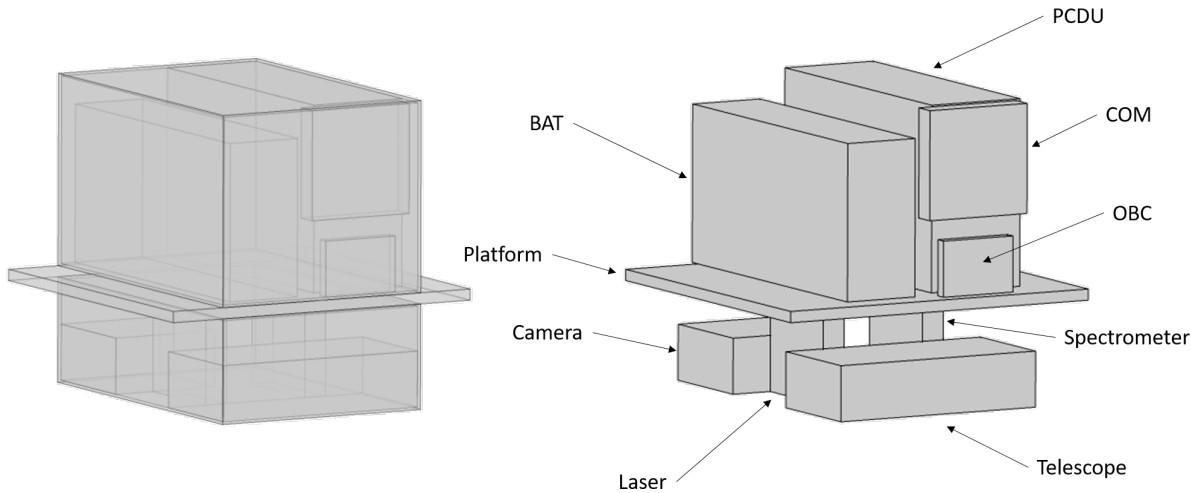

**Figure 6.** Detailed views of the electronics and instrument compartment of the probe, showing dummy volumes for the subsystems (housing not shown on right image).

**Table 2.** System power and mass budget.

| Device/Instrument | Power [W] (Added Individual Margin) | Mass [kg] (Added Individual Margin) |
|---|:---:|:---:|
| **Science payload 1** Payload Raman (UV laser/Telescope/Spectrometer) | 55.0 (10%) | 1.7 (5%) |
| **Science payload 2** High-resolution camera (Imperx B3412) | 5.4 (5%) | <0.3 |
| Temperature sensor (PT1000) | 0.1 (5%) | - |
| Humidity sensor (SHTW2) | 0.1 (5%) | - |
| Wind/Pressure sensor | 0.1 (5%) | - |
| **Power Subsystem** PCDU | 16.7 (10%) | 2.9 (10%) |
| Li-SOCl$_2$ battery cell, LSH 20-150, 8s4p module | - | 4.2 (5%) |
| **Command and Data Handling** OBC (LEON 3, ProASIC3) | 2.2 (10%) | 0.1 (10%) |
| **Communications** S-band antenna module (NanoCom ANT2000) | 11.6 (5%) | 0.1 (10%) |
| **Balloon/deployment system** Floating mass (gas + balloon) | - | 5 (10%) |
| **TOTAL (including margin)** | **91.1 (109.3) W** | **14.3 (17.6) kg** |

Inspiration for the presented design came from the Vega missions, which carried a payload of 6.9 kg, using a 3.5 m diameter balloon (total floating mass of 21 kg). The Vega balloon was made of a heavy high temperature resistant Teflon cloth matrix, coated with Teflon film. The recent developments in material science allow for using lightweight flexible composites, for example engineering films such as Mylar, Nylon, Kapton, PBO, or Tedlar, reinforced with high strength fibres such as Vectran, Zylon, or Kevlar [50]. The working gas for Vega 2 was helium, with a diffusion rate through the Teflon fabric slow enough that the balloon outlived the battery lifetime of 48 h. PBO or coated Mylar not only show slow gas permeability, but are also lightweight [51]. This, together with the proposed power system design (Section 3.2), should allow for a minimum lifetime of two weeks.

The gondola on Vega 2 was painted with a white reflective coating to protect it from the corrosive sulphuric atmosphere and reflect the surface albedo. Previous drafted missions, such as VALOR+, also suggested the use of Teflon coated balloons for better sulphuric acid resistance [52]. As a modification to the floating mass design in comparison to Vega 2, it would be preferred to manufacture the payload housing from a carbon–carbon material (density of 1.9 g/cm$^3$), painted with the highly emissive black coating "Solar Black", developed for the ESA Solar orbiter mission [53]. A similar option was also planned for the ESA missions to Titan and Enceladus [54].

*3.1. Science Optical Payload*

The main scientific payload is designed to enable both Raman and fluorescence analysis; moreover, the planned instrument will be implemented using a time-gated detector to differentiate both signals. Both processes can be stimulated using a single laser, while the response signals are analysed by a single spectrometer. Time-resolved measurements should be included to differentiate the raised signal coming from the Raman shift or from the scattered fluorescence effect. In the case of the fluorescence signal, being of higher intensity, it could be used to acquire information in the tens of metres distance range across the atmosphere [23]. A similar approach is planned to be used in the OrganiCam instrument from Los Alamos National Laboratory (Los Alamos, New Mexico, United

States), in a mission to Jupiter's moon Europa. The mission plans to use a combination of fluorescence/Raman instruments to search for organic material [55]. Thus, the following instrument baseline architecture is suggested (see Figure 7) [24]:

- The UV laser source is separated using beam-splitters to transport the emitted light in free space into the different regions of study:
  – One of the beams is directed into a small probe side cavity (a few mm). Across the cavity, designed to stabilise the atmosphere, the device performs transmission Raman. On the opposite side of the cavity, the receiving optics guide the incoming light into the spectrometer. The instrument would be able to provide a more accurate study of single proximity molecules and possible amino acids.
  – The other beam is directed through an aperture into the atmosphere. The laser illuminates a few tens of metres and the fluorescence signal is collected with a small telescope (around 100 mm aperture).

- Both fluorescence and Raman signals are joined using a beam combiner. The incoming signals are independently analysed using a single spectrometer. The Raman signal is used to study the chemical bonds and possible functional groups of the molecules, while the fluorescence signal is focused on electronic structure to analyse aromatics and aldehydes [56].

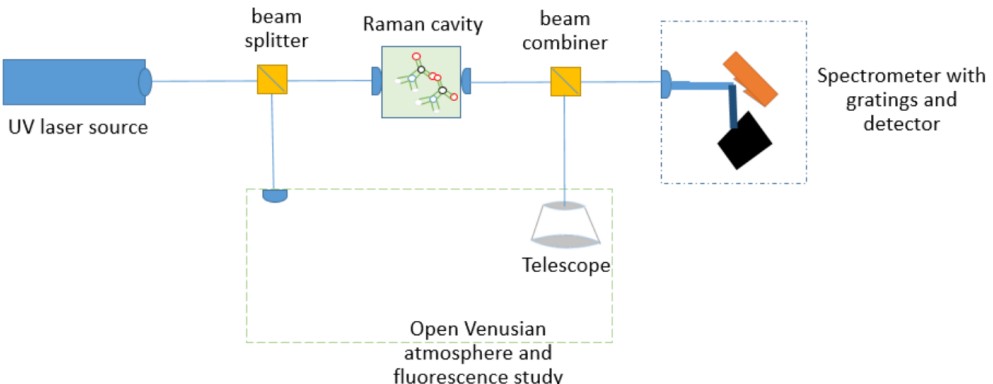

**Figure 7.** Schematic of the instrument baseline. The laser is obliquely placed near the receiving telescope.

The laser source and sensitive spectrometer parts need to be protected from the extreme environment and therefore accommodated inside the instrument compartment of the probe.

Due to the temperature ranges of the atmosphere and the interest in studying the presence of amino acids and their response to certain wavelengths, the proposed excitation source is in the UV region, where these molecules show great absorption and fluorescence. UV lasers have recently been market accessible [57]. Even if, nowadays, reliable UV lasers can perform Raman spectroscopy in a laboratory environment, the laser qualification for space missions is not yet a straightforward and standardised process [58]. However, there are examples of space-qualified UV laser spectrometer devices, such as the SHERLOC instrument of the Mars 2020 mission [27] or the UV-DPSS of the MOMA (Mars Organic Molecule Analyser) instrument on the ExoMars 2022 mission [59].

Different instrument options are available for this mission, however, the best solution will always be a trade-off between science capabilities (different sample analysis and used methods) and the mission constraints (budgets and environmental conditions). Due to the principal objective of studying larger molecules, plus budget constraints and space heritage, the preferred baseline is an NeCu hollow cathode ion laser, emitting at 248 nm wavelength [60], similar to the one used for the SHERLOC instrument [61]. Other UV configurations, for instance changing the hollow cathode ion laser to a frequency dou-

bled solid state laser, or even tunable lasers with a wider emission wavelength [62], are possible alternatives.

Within this mission concept, it is proposed to perform a time-resolved Raman analysis [23] by using an intensified charge-coupled device (ICCD) time gated measurement detector, to better separate the Raman and fluorescence signals.

Additionally, the science payload includes a high definition camera for the visualisation of the analysed atmosphere region. Miniaturised COTS cameras for such purposes have been widely implemented in space missions over the recent decades [63]. For the proposed concept, the HiREV2 camera [64] was selected.

### 3.2. Power

The power subsystem architecture represents one of the major design trade-offs. Using an architecture to manage either non-rechargeable battery cells or rechargeable cells has a considerable effect on the mission performance, especially regarding the system mass and complexity, as well as the mission duration. The first option (non-rechargeable cells) was selected as the baseline, while the second one (rechargeable cells) was studied as an alternative. Both options are analysed in the following sections.

### 3.2.1. Option 1: Non-Rechargeable Cells, No Solar Arrays

Non-rechargeable battery cells show a superior power density compared to rechargeable ones. Their main drawback is that, in contrast to a solar-array-based power subsystem, the mission will be terminated as soon as the batteries are depleted. On the other hand, the electrical power system is simpler, since no solar array regulators (SARs) are needed, complemented by a simplified structure of the probe that does not require any deployment mechanism.

Battery Sizing

Several battery cells have been investigated, focusing on lithium-based $Li-SOCl_2$ and $Li-SO_2$, given their outstanding energy density [65]. Table 3 shows their main characteristics. It can be seen that the LSH20-150 cell offers a distinct extended operating temperature range, which could be interesting for exploring a wider atmosphere area. On the other hand, the LS33600 cells show the highest energy density, leading to a lower system mass.

**Table 3.** Characteristics of investigated non-rechargeable battery cells.

| Cell name | LS14250 | LS33600 | LSH20 | **LSH20-150** | LO34SX | LO25SX | LO 29 SHX | LO 39 SHX | G 06/6 | G 62/1 |
|---|---|---|---|---|---|---|---|---|---|---|
| Type | $Li-SOCl_2$ | $Li-SOCl_2$ | $Li-SOCl_2$ | $Li-SOCl_2$ | $Li-SCl_2$ | $Li-SCl_2$ | $Li-SCl_2$ | $Li-SCl_2$ | $Li-SCl_2$ | $Li-SCl_2$ |
| Energy [Wh] | 4.3 | 61.2 | 46.8 | 50.4 | 2.8 | 22.4 | 10.5 | 32.2 | 2.7 | 95.2 |
| Typical weight [g] | 8.9 | 90 | 100 | 104.5 | 16 | 96 | 40 | 125 | 15 | 300 |
| Operating temp. [°C] | −60/+85 | −60/+85 | −60/+85 | −40/+150 | −40/+70 | −60/+70 | −60/+70 | −60/+70 | −60/+70 | −60/+70 |
| Energy density [Wh/kg] | 485.4 | 680.0 | 468.0 | 482.3 | 175.0 | 233.3 | 262.5 | 257.6 | 177.3 | 317.3 |

The mission requirement TR6 states that the system shall be operating for at least two weeks. The instrument measurements can be performed in a few seconds and repeated with a certain frequency over that period of two weeks. There is a compromise between the available energy and the mass of the system. Therefore, it was assumed that the battery should provide enough energy storage to operate for a total time of 12 h, distributed over the nominal mission duration, at the maximum power level of 109.3 W as stated in Table 3. With this assumption, the required energy is 1.3 kWh. Table 4 summarises the required battery module configuration (number of cells in a series, assuming a 28 V bus voltage and cells in parallel, to provide enough energy and discharge capability) and the corresponding mass for all the analysed non-rechargeable battery cells.

**Table 4.** Required number of non-rechargeable cells and corresponding mass, excluding the module mechanical assembly.

| Cell name | LS14250 | LS33600 | LSH20 | **LSH20-150** | LO34SX | LO25SX | LO 29 SHX | LO 39 SHX | G 06/6 | G 62/1 |
|---|---|---|---|---|---|---|---|---|---|---|
| Type | Li-SOCl$_2$ | Li-SOCl$_2$ | Li-SOCl$_2$ | Li-SOCl$_2$ | Li-SCl$_2$ | Li-SCl$_2$ | Li-SCl$_2$ | Li-SCl$_2$ | Li-SCl$_2$ | Li-SCl$_2$ |
| Series cells | 8 | 8 | 8 | 8 | 10 | 10 | 10 | 10 | 10 | 10 |
| Parallel cells | 38 | 3 | 4 | 4 | 47 | 6 | 13 | 5 | 50 | 2 |
| Cells mass [kg] | 2.7 | 2.2 | 3.2 | 3.3 | 7.5 | 5.8 | 5.2 | 6.3 | 7.5 | 6 |
| Total mass [kg] | 3.2 | 2.6 | 3.8 | 4.0 | 9.0 | 6.9 | 6.2 | 7.5 | 9 | 7.2 |

The option with LS33600 cells appears to be the most suitable option in terms of mass. However, for the sake of extending the possible altitude range of the probe with respect to ambient temperatures, the battery module with LSH20-150 cells is preferable. Furthermore, the mass of the LSH20-150 modules is still in a feasible range. Hence, the selected battery module uses LSH20-150 cells, in a configuration of 8s4p (eight cells in series, four in parallel), with a total assembly mass of 4 kg.

Architecture

There are two options regarding the bus voltage: regulated or unregulated. For the first case, only a Battery Discharge Regulator (BDR) is required to keep the bus voltage at 28 V. This BDR must include overdischarge protections, maximum current discharge control and high output dynamics. For the second case, the batteries would determine the bus voltage, leading to a voltage range from 16.8 V to 29.4 V with the LSH20-150 battery. Nevertheless, overdischarge protections must also be included in this option. Given the sensitivity of the instrument, a regulated bus voltage is preferable. The Power Conditioning and Distribution Unit (PCDU) must also include a dedicated power supply for the UV laser [46].

3.2.2. Option 2: Rechargeable Cells and Solar Arrays

Despite the adverse environmental conditions of Venus' atmosphere, a solution with solar cells might potentially be feasible [66]. Adding solar arrays to the probe increases the system complexity, since SARs are required to condition the power they provide, as is a BCDR module to regulate the charge and discharge of the batteries. Nevertheless, this design option provides continuous electrical power generation during daytime, allowing to recharge the batteries and hence enabling a much longer mission duration. On the other hand, the operating temperature of these batteries is more restrictive, which would limit the altitude range that can be explored. Hence, this option is analysed just as a secondary alternative.

The batteries are considered as a backup electrical energy storage, either to complement the solar arrays to provide some extra peak power or to provide some energy in survival mode (i.e., when the solar arrays are not capable of providing enough power for the platform).

Battery Sizing

Despite the lower energy density of rechargeable battery cells compared to non-rechargeable ones, they will be recharged by the solar arrays, ultimately leading to a longer lifetime. Nevertheless, the measurements shall be duty-cycled in order to keep a positive energy balance.

Table 5 shows the main characteristics of several Lithium-ion cells [65]. Despite showing the highest energy density, the maximum operating temperature of the MP 176065 xlr cells is constrained to 60 °C. For this mission, MP 176065 xtd cells are more suitable, since they can be operated up to 85 °C and the energy density is close to that of the previous cells.

**Table 5.** Characteristics of investigated rechargeable battery cells.

| Cell name | MP 174565 XTD | **MP 176065 XTD** | VL 34570 xlr | MP 176065 xlr | VES16 |
|---|---|---|---|---|---|
| Type | Li-ion | Li-ion | Li-ion | Li-ion | Li-ion |
| Life cycles @100DoD, C-C/2, 25 °C | 2700 | 2700 | 600 | 1800 | 5000 |
| Energy [Wh] | 14.6 | 20.4 | 19.7 | 24.8 | 16 |
| Typical weight [g] | 97 | 135 | 130 | 150 | 155 |
| Discharging temp. [°C] | −40/+85 | −40/+85 | −35/+60 | −35/+60 | +10/+40 |
| Charging temp. [°C] | −30/+85 | −30/+85 | −30/+60 | −30/+60 | +10/+40 |
| Energy density [Wh/kg] | 150.5 | 151.4 | 151.6 | 165.5 | 103.2 |

As mentioned in the previous section, mission objective TR6 establishes that the lifetime of the probe should be at least two weeks. The solar cells will provide continuous power during the daytime, although the effective solar flux diminishes with lower altitudes. Hence, the batteries should be sized to provide at least 2 h of continuous discharge at the average power from Table 5 (109.3 W, 2 h, 80% DoD, 90% retained energy at EOL = 303.6 Wh nameplate energy) and also to provide at least enough energy for a pulse with the peak power (130 W, 60 μs = 7.8 mWh ). Hence, batteries must have a nameplate energy higher than 303.6 Wh. This way, once the batteries reach 80% DoD, the payload would remain inactive until the batteries are recharged again if the solar arrays are receiving sunlight. Otherwise, if the probe is in eclipse condition after those two hours, the whole platform would remain dormant until sunlight is received again so that the batteries will start being recharged again.

Considering a DoD of 80% would lead to a degradation below 90% at EOL (assumed 2 months, depending on the descending speed, hence on the temperature and the frequency of the measurements too), approximately. Applying these factors, the required mass of battery cells is shown in Table 6 for different cells.

**Table 6.** Required number of rechargeable battery cells (series and parallel), cells' mass and total mass of the battery module (including the mechanical assembly and electrical connectors).

| Cell name | MP 174565 XTD | **MP 176065 XTD** | VL 34570 xlr | MP 176065 xlr | VES16 |
|---|---|---|---|---|---|
| Type | Li-ion | Li-ion | Li-ion | Li-ion | Li-ion |
| Series cells | 8 | 8 | 8 | 8 | 8 |
| Parallel cells | 3 | 2 | 2 | 3 | 3 |
| Cells mass [kg] | 2.3 | 2.2 | 2.1 | 2.4 | 3.7 |
| Total mass [kg] | 2.8 | 2.6 | 2.5 | 2.9 | 4.5 |

MP 176065 xtd cells are selected as an alternative battery for this mission due to their wide operating temperature range. Since its nominal voltage is 3.65 V, the configuration of the battery module must be 8s2p (eight cells in series, two in parallel). With this configuration, the mass of the battery only is 2.4 kg and 2.9 kg after adding the mass of the electrical connectors and the mechanical assembly.

Solar Array Sizing

There are different factors that influence the performance of the solar cells in the atmosphere. The absorption and scattering of the light depends on the altitude, and is mainly affected by the thick main cloud layer (48 km to 65 km). Furthermore, the temperature increases with lower altitudes, hence reducing the efficiency of the solar cells. The generated power is also dependent on the Sun angle, therefore the equator is the most favourable scenario in terms of power due to a higher angle of incidence. Furthermore, the clouds contain corrosive components, so a special coating would be required for the solar arrays.

As mentioned in Section 1.4, the region of interest is the equator, targeting altitudes from 40 km to 70 km, although, as will be shown in the next section, altitudes below 55 km might not be feasible from a thermal perspective. The corresponding power generated per

unit area of triple-junction solar cells is 112.2 W/m$^2$, 256 W/m$^2$ and 700 W/m$^2$ for 40 km, 55 km and 70 km, respectively [66].

The simplest option would be mounting as many cells as possible on the top surface of the probe, whose size is 300 × 200 mm (see Figure 5). Assuming a packing factor of 85%, it is possible to mount 16 Azurspace (AZUR SPACE Solar Power GmbH, Heilbronn , Germany) 3G30 cells, leading to 0.048 m$^2$ of cell surface.

Another option is adding deployed panels to expand the solar array (SA) surface, at the expense of increasing the mass of the probe and the complexity of the mechanical structure of the probe and the additionally required deployment mechanism. In this case, there would be two 300 × 200 mm panels in addition to the previous body mounted. In other words, the solar cell surface would be triplicated, leading to 0.145 m$^2$.

Lastly, in order to further increase the power generated, these deployed SA can be doubled. As a consequence, the solar cells' surface is multiplied by five compared to the body-mounted option.

Table 7 summarises the power generated with these two options depending on the altitudes of interest.

**Table 7.** Power generated by different solar array configurations at different altitudes.

| Altitude (power per unit area) | 40 km (112.2 W/m$^2$) | 55 km (256 W/m$^2$) | 70 km (700 W/m$^2$) |
|---|---|---|---|
| 1 body mounted SA (16 cells = 0.048 m$^2$) Power [W] | 5.4 | 12.3 | 33.7 |
| 1 body mounted + 2 deployed SA = 48 cells = 0.145 m$^2$,Power [W] | 16.2 | 37.0 | 101.2 |
| 1 body mounted + 4 deployed SA = 80 cells = 0.241 m$^2$,Power [W] | 27.0 | 61.7 | 168.7 |

Operating at 40 km should be regarded as a critical case, due to thermal constraints. On the other hand, at 55 km, none of the options are capable of providing enough power to operate the complete system continuously with only the power from the SA. Therefore, the SA must be used only to charge the batteries, but not to supply all subsystems in a continuous manner. Instead, instruments and communications shall be duty-cycled in order to ensure a positive energy balance.

With the aim of increasing the reliability of the system, the option with a body mounted panel is more suitable and the mass increase due to this SA is just 400 g. However, the main drawback is the long time required to recharge the batteries (about 9 h, at 55 km altitude, to recharge after 1 h of operations with all subsystems active).

The other options with two or four deployed SA would increase the mass by 1.2 kg and 2.4 kg on top of that. In addition, these options would require a deployment mechanism, which might introduce some risk—the mission would be compromised if this mechanism fails.

Architecture

As mentioned above, a regulated bus is preferable given the sensitivity of the instrument. However, in this case, not only is a BDR required, but also a BCDR and an SAR. The SAR can be implemented either using Maximum Power Point Tracking (MPPT) or a Sequential Switching Shunt Regulator (S3R). MPPT is more suitable, since it is able to maximise the power that can be generated given the variable environmental conditions of this mission. There are some commercial EPS compatible with these features, such as [67,68], although an additional specific power supply for the UV laser must be included in the PCDU.

3.2.3. Selected Option and Performance

Given the characteristics of the mission and with the aim of maximising the reliability while minimising the system mass and complexity, the selected power subsystem consists of non-rechargeable cells (in particular, a battery module with LSH20-150 cells, with a 8s4p configuration) and a PCDU, which contains a BDR to regulate the bus voltage to 28 V and

a dedicated power supply for the instrument's laser. With this configuration, the target of a two week lifetime is achieved by duty-cycling the instruments.

### 3.3. Thermal

Following the mission objective TR1 (see Section 2.1), the probe should operate at an altitude between 40 and 70 km. The estimated ambient conditions in this altitude region are temperatures of 225–400 K (−48 to 127 °C), pressures on the order of 10–1000 mbar and wind speeds around 40–100 m/s. Temperature profiles measured by earlier missions, such as Vega-2 and the Pioneer Venus probes, indicate a very low temporal and spatial variability on the order of 5 K at a given pressure or altitude level [33,69].

The main subsystems of the probe impose thermal requirements on the design, as summarised in Table 8. It is clear from these requirements that the upper end of the expected temperature range will be more challenging for the thermal design than the lower temperatures. High-performance insulation can be utilised to protect the probe from the hot atmosphere. Aerogels, multi-layer-insulation (MLI) and phase-changing materials (PCM) have previously been identified as relevant technologies to extend the lifetime of Venus probes [70]. However, it needs to be considered that not all of these materials might be qualified for high-temperature or high-g entries. The efficiency of porous insulators like aerogel and MLI also heavily depends on gas pressure, hence they typically find their application in the free molecular flow regime. Due to the ambient pressure of approximately 10–1000 mbar in the targeted altitude range, passive cooling and heating is mainly achieved via convection. Active cooling options include heat pipes and miniaturised cryo-coolers, but these require additional mass and power and only provide limited cooling capability on this small scale.

**Table 8.** Temperature requirements (non-operational and operational) and estimated heat dissipation of all subsystems.

| Subsystem | Non-Op Min [°C] | Op Min [°C] | Op Max [°C] | Non-Op Max [°C] | Heat Dissipation [W] |
|---|---|---|---|---|---|
| UV laser [a] | n/a | −135 | 70 | n/a | 3.4 |
| Spectrometer | −60 | −30 | 50 | 70 | 8.2 |
| Telescope | −60 | −30 | 50 | 70 | 3.4 |
| Camera [b] | −40 | −30 | 60 | 85 | 1.0 |
| BAT [c] | n/a | −40 | 150 | n/a | 0.6 |
| PCDU [d] | −40 | −30 | 60 | 85 | 2.0–10.3 |
| OBC | −55 | −30 | 85 | 125 | 0.4 |
| COM [e] | n/a | −40 | 70 | n/a | 2.2 |

[a] Photon Systems NeCu laser, [b] Imperx B3412, [c] Saft LSH 20-150 (8s4p), [d] incl. laser power supply, [e] Gomspace ANT2000.

Due to the atmospheric gas pressure and wind speeds, the temperature at the external surfaces is less sensitive to the orientation of the probe. Table 9 provides an overview of the estimated flow conditions in the atmosphere at different altitudes, with associated ambient temperature and pressure (compare Figure 3). The convective heat exchange coefficient over a plate with length 300 mm, which equals the length of the probe's external surface, was calculated for a wind speed of 40–100 m/s, covering the relevant range at the expected altitudes. The actual relative wind speed between the probe and the ambient atmosphere is difficult to estimate at this stage, since this requires knowledge of the probe–balloon dynamics and the relative motions involved. As Table 9 shows, the increasing temperature and pressure at lower altitudes lead to an increasingly turbulent flow and thus to higher heat transfer on the external surfaces of the probe. This further complicates maintaining the operational temperatures within their limits at lower altitudes. Furthermore, in the lower cloud (ca. 46–55 km), the probe will experience stronger updraft and downdraft on the order of 2–3 m/s, which additionally contributes to a stronger convective heat exchange [71]. Assuming an absolute worst case with no relative motion between the atmosphere and the probe (and also neglecting natural convection), the maximum steady state temperatures are 48 °C at 70 km, 136 °C at 55 km, and 223 °C at 40 km, and therefore

clearly exceed the operational range. However, such a case is rather unlikely, as there will always be convection to some extent.

**Table 9.** Estimated ambient conditions and calculated convective heat transfer properties for a 300 mm long plate. The value range corresponds to wind speeds between 40 m/s and 100 m/s.

| Altitude | Temperature | Pressure | Reynolds Number | Nusselt Number | Heat Exchange Coefficient | Max. Steady State Temp. (Science Payload Off) |
|---|---|---|---|---|---|---|
| 70 km | −48 °C | 10 mbar | $1.9 \times 10^4$–$4.8 \times 10^4$ (laminar) | 43–68 | 2–4 W/(m² K) | −45 … −43 °C |
| 55 km | 40 °C | 100 mbar | $1.4 \times 10^5$–$3.4 \times 10^5$ (laminar) | 115–182 | 6–10 W/(m² K) | 41 … 42 °C |
| | | | $1.4 \times 10^5$–$3.4 \times 10^5$ (turbulent) | 356–741 | 20–41 W/(m² K) | 40 … 41 °C |
| 40 km | 127 °C | 1000 mbar | $1.1 \times 10^6$–$2.7 \times 10^6$ (turbulent) | 1509–4004 | 84–223 W/(m² K) | 127 °C |

The preliminary thermal analysis showed that the steady state temperatures of the components listed in Table 8 are only a few degrees above the ambient temperature for different boundary conditions (Table 9). There are only very small thermal gradients between the subsystems, because the inside of the probe is at ambient pressure. The results suggest that the required operational temperatures can be maintained, at least in parts of the targeted altitudes (55–70 km), representing the upper and middle cloud layer. Descending further down, the minimum temperatures will ultimately reach the ambient value (127 °C at 40 km) in the absence of an active cooling mechanism. For operation at and above 70 km, with the science payload being switched off, the probe requires additional heaters to maintain a safe margin to the minimum temperatures of all subsystems. With appropriate insulation of the science payload, short excursions into cloud layers below 55 km might in principle be possible, assuming a colder initial temperature when starting the descent.

To further investigate the transient thermal behaviour, a baseline operational mode was applied to the science payload (UV laser, spectrometer, telescope and camera) with a duty cycle of 60 s on, followed by 540 s off, so that a measurement is performed every 10 min. The PCDU, onboard computer, as well as the communication subsystem and the sensors, are constantly active during operations. The resulting temperature evolution is depicted in Figure 8 for an altitude of 70 km (−48 °C and 10 mbar ambient) and 55 km (40 °C and 100 mbar ambient). The proposed duty cycle appears to be appropriate for limiting the temperature increase during nominal operation to a few degrees, while providing sufficient time for a full measurement with the spectrometer and imaging with the camera. However, the presented thermal analysis is only preliminary and more details on the placement of components and their material properties will have to be taken into account for a more conclusive analysis.

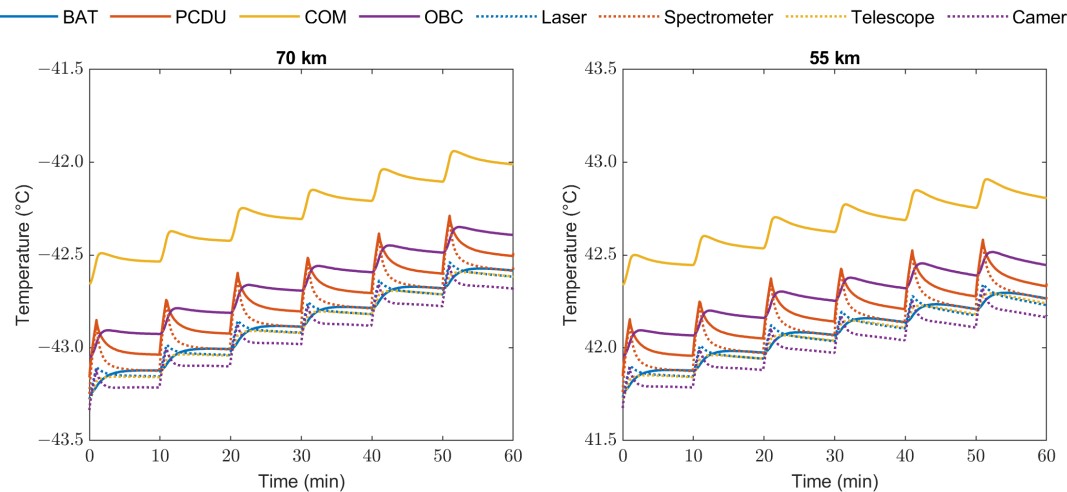

**Figure 8.** Transient heating of the subsystems with baseline duty cycle of laser, spectrometer, telescope and camera. The initial temperature equals the steady state temperatures with the aforementioned components switched off.

## 4. Conclusions

In the present work, a conceptual design for a small probe to study the atmoshpere of Venus is proposed. The science objectives were analysed and it was shown that the mission can be realised using existing COTS components (already used in previous space missions, mainly LEO) that favour a lower total cost. Therefore, the indicated technical requirements, TR3, TR4, and TR5 (see list of mission objectives in Section 2.1), can be satisfied.

The main technical challenge for the probe is the thermal control at lower altitudes, where ambient temperature and pressure complicate the cooling of the internal components. A preliminary analysis showed that, without active cooling mechanisms, the lowest operational altitude of the probe should be approximately 55 km, where the temperature is around 40 °C. While it is recommended to use a science payload with flight heritage, such as the SHERLOC instrument on board the Mars 2020 rover, it is also strongly recommended to put effort into qualifying such instruments already available in Europe for operation in space and planetary environments.

Here, a preliminary design is presented, based on the scientific payload using UV and Raman spectroscopy together with a visual camera and a complementary sensor suite. However, the current design is not limited to these instruments, and accommodation of others, such as a nephelometer, might be a valuable addition. With the current scientific instruments baseline, the mission will fulfill the scientific objective TR2. Furthermore, as a detailed power system trade-off showed, the mission will be able to operate for several days across different atmospheric regions, and thus satisfy the operational objective TR6.

**Author Contributions:** All authors contributed equally. All authors have read and agreed to the published version of the manuscript.

**Funding:** This research received no external funding.

**Institutional Review Board Statement:** Not applicable.

**Informed Consent Statement:** Not applicable.

**Data Availability Statement:** Not applicable.

**Acknowledgments:** The authors want to thank Antonio Sansano-Caramazana, Helene Strese and Luca Maresi, co-authors of a previous conference publication (ICSO2021), which inspired the idea for the present article.

**Conflicts of Interest:** The authors declare no conflict of interest.

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
