# Peer review of "Exploring the Habitability of Venus: Conceptual Design of a Small Atmospheric Probe"

_aerospace, doi:10.3390/aerospace8070173_

Round 1
Reviewer 1 Report
Overall, this is well-presented mission concept based on a simple, yet interesting idea. Based on my expertise on Venus atmosphere, I have comments on wind conditions and thermal design (that should be easily adressed) and several small remarks.
Main comments:
The assumption of 100 m/s for the wind values are true for the cloud top only. At lower altitudes the zonal wind decreases, with a linear trend – to the first order – from 100 m/s to 40 m/s between 70 km and 40 km (see for instance the Pioneer Venus profiles, e.g. Read and Lebonnois, their figure 1).
Figure 1 from Read, P. L., & Lebonnois, S. (2018). Superrotation on Venus, on Titan, and elsewhere. Annual Review of Earth and Planetary Sciences, 46, 175-202. https://doi.org/10.1146/annurev-earth-082517-010137
Is the assumption that zonal wind are 100 m/s in the 40-70 km altitude range an assumption you made as an upper limit scenario for engineering design purposes? In that case you should state that point explicitly. If not, please take into account the fact that wind speed is lower at lower altitudes in your analysis.
Then the sole purpose of the wind speed values is for the thermal analysis in section 3.3 and I am not sure to follow your point here. The wind is a background condition, and the atmospheric probe would simply be carried away by the large-scale winds, without experiencing local wind of 100 m/s, unless there are strong gusts, but nothing indicate gusts could reach 100 m/s inside Venus clouds. The reasoning behind table 9 is invalid as far as I can tell.
Section 3.2.2: The circumnavigation period in Venus clouds goes from 4 days at the cloud top to 9 days at the lower cloud layer. Given the two weeks mission duration, you need to take into account night passes. How does the rechargeable option takes into account this fact? Is two weeks the total mission length? Or just the sum of the active parts?
I would also bring to your attention that the lower cloud (48-52 km) is highly convective, with strong updraft and downdraft. For instance see Lefèvre 2018 https://doi.org/10.1029/2018JE005679 . Diving into that layer would require some specific design, well beyond the purpose on that article.
Other remarks:
L34 What does atmosphere disequilibrium mean?
L35-36 Please be less assertive in your formulation with lifeforms being a reason for atmospheric observations
Now that 3 Venusian missions have been selected since your initial submission (!!!!), you may want to reformulate a bit L48, section 2.2, etc …
L74: give due credit and precise that Akatsuki is also an atmospheric mission, but from the orbit.
Figure 1: The lack of information is confusing and it took me some time to understand this figure as a non-specialist. Add some explanation in the figure or its caption, explaining that each line “excites” a given spectral region. Also there is a typo: “wavelenght”
L169: change N for N2
L363 typo “Archtiecture”
L485-486: Use explicit values when you qualify “dense” and “lower temperatures” conditions. Is it compared to room conditions?
Reviewer 2 Report
The article titled "Exploring the Habitability of Venus: Conceptual Design of a Small Atmospheric Probe" describes the conceptual design of a probe devoted to studying the Venus atmosphere to search for, among other compounds, biosignatures. The Scientific objectives are well delineated in the Introduction with plenty of references. The design of the probe is also well outlined considering the main risks and their mitigation. The paper is written in fluent English in a really clear way. I've no Major concerns, just one Minor:
- Figure2: it is difficult to read the labels and the numbers, please make a picture with higher resolution.
From my point of view, the paper could be published as it is.
Reviewer 3 Report
Ribes-Pleguezuelo et al. Comments
Thank you very much for letting me review your paper, the work presented seems to be novel and definitely has an interesting application.
Major general comments:
The subject of the paper is very interesting and suitable for publication. I will admit that this is outside my expertise, but this paper is well written, well organized, and presented. This work describes a probe to be sent along side another mission to Venus in order to understand the composition in ways other Venetian probes have not been able to in the past. The Raman spectrometer will be able to detect organics including amino acids, phosphine, and even potentially verify the presence of biosignatures.
The work needed to make the project feasible is described in detail in the Probe Design section. I will say that perhaps it would be a good idea to make a Table 10 that shows a summary of what needs to be done in order to prepare the mission along with a timeline. Sort of like a small recap of section 3 but in tabular form.
Another optional suggestion is to mark on Figure 3 where the probe could fly in the Venetian atmosphere without heating or cooling elements, the 55-70 km range. I would also suggest marking where the probe could fly with heating and cooling elements, but I would assume this second part may be a little more ambiguous so if it is not possible then skip it.
I would suggest to carefully go through the manuscript and correct some grammatical errors. I found a few throughout of which some I noted in the “Line by line minor points” section.
Overall, this paper is great, and I think it will make a great contribution. Hopefully, this concept gets turned into a real mission!
Minor general comments:
Figure 2 seems a little low in resolution. Please increase the resolution if possible.
Table 1, change “Dioxid” to “Dioxide”
Table 6, is the number of parallel cells for “MP 174565 XTD” supposed to be 32?
Line by line minor points:
11: “The current state-of-the-art of the required technologies…” does not make sense. Maybe you meant to say, “Current state-of-the-art technologies would allow a more…”. Please reword.
20: “an” to “a”
23: “is” to “are”
25: “is” to “are”
91: “micrometre” to “micrometer”
100: delete the word “Already”
156: replace “and on ground” to “and on the ground”
156: “degree” to “degrees”
157: “degree” to “degrees”
163: should “VeGA” be “Vega”?
168: delete “or not”
201: delete “of the”
227: “Figure” to “Figures”
273: “is” to “are”
399: replace “during daytime” to “during the daytime”
430: replace “m2” to “m2”
476: should “VeGa-2” be “Vega-2”?
